# MANDERA: Malicious Node Detection in Federated Learning via Ranking

## Abstract

Byzantine attacks hinder the deployment of federated learning algorithms. Although we know that the benign gradients and Byzantine attacked gradients are distributed differently, to detect the malicious gradients is challenging due to (1) the gradient is high-dimensional and each dimension has its unique distribution and (2) the benign gradients and the attacked gradients are always mixed (two-sample test methods cannot apply directly). To address the above, for the first time, we propose MANDERA which is theoretically guaranteed to efficiently detect all malicious gradients under Byzantine attacks with no prior knowledge or history about the number of attacked nodes. More specifically, we transfer the original updating gradient space into a ranking matrix. By such an operation, the scales of different dimensions of the gradients in the ranking space become identical. The high-dimensional benign gradients and the malicious gradients can be easily separated. The effectiveness of MANDERA is further confirmed by experimentation on *four* Byzantine attack implementations (Gaussian, Zero Gradient, Sign Flipping, Shifted Mean), comparing with state-of-the-art defenses. The experiments cover both IID and Non-IID datasets.

## 1 Introduction

Federated Learning (FL) is a decentralized learning framework that allows multiple participating nodes to learn on a local collection of training data. The updating gradient values of each respective node are sent to a global coordinator for aggregation. The global model collectively learns from each of these individual nodes by aggregating the gradient updates before relaying the updated global model back to the participating nodes. The aggregation of multiple nodes allows the model to learn from a larger dataset which will result in the model having greater performance than if each node was to learn on their local subset of data. FL presents two key advantages: (1) the increase of privacy for the contributing node as local data is not communicated to the global coordinator, and (2) a reduction in computation by the global node as the computation is offloaded to contributing nodes.

However, FL is vulnerable to various attacks, including data poisoning attacks Tolpegin et al. (2020) and Byzantine attacks Lamport et al. (2019). The presence of malicious actors in the collaborative process may seek to poison the performance of the global model, to reduce the output performance of the model Chen et al. (2017); Baruch et al. (2019); Fang et al. (2020); Tolpegin et al. (2020), or to embed hidden back-doors within the model Bagdasaryan et al. (2020). Byzantine attack aims to devastate the performance of the global model by manipulating the gradient values. These gradient values that have been manipulated are sent from malicious nodes which are unknown to the global node. The Byzantine attacks can results in a global model which produces an undesirable outcome Lamport et al. (2019).

Researchers seek to defend FL from the negative impacts of these attacks. This can be done by either identifying the malicious nodes or making the global model more robust to these types of attacks. In our paper, we focus on identifying the malicious nodes to exclude the nodes which are deemed to be malicious in the aggregation step to mitigate the impact of malicious nodes. Most of the existing methods rely on the gradient values to determine whether a node is malicious or not, for example, Blanchard et al. (2017); Yin et al. (2018); Guerraoui et al. (2018); Li et al. (2020); Fang et al. (2020); Cao et al. (2020); Wu et al. (2020b); Xie et al. (2019; 2020); Cao et al. (2021) and So et al. (2021). All the above methods are effective in certain scenarios.

There is a lack of theoretical guarantee to detect all the malicious nodes in the literature. Although the extreme malicious gradients can be excluded by the above approaches, some malicious nodes could be mis-classified as benign nodes and vice versa. The challenging issues in the community are caused by the following two phenomena: [F1] the gradient values of benign nodes and malicious nodes are often non-distinguishable; [F2] the gradient matrix is always high-dimensional (large column numbers) and each dimension follows its unique distribution. The phenomenon [F1] indicates that it is not reliable to detect the malicious nodes only using a single column from the gradient matrix. And the phenomenon [F2] hinders us from utilizing all the columns of the gradient matrix, because it requires a scientific way to accommodate a large number of columns which are distributed considerably differently.

In this paper, we propose to resolve these critical challenges from a novel perspective. Instead of working on the node updates directly, we propose to extract information about malicious nodes indirectly by transforming the node updates from numeric gradient values to the ranking space. Compared to the original numeric gradient values, whose distribution is difficult to model, the rankings are much easier to handle both theoretically and practically. Moreover, as rankings are scale-free, we no longer need to worry about the scale difference across different dimensions. We proved under mild conditions that the first two moments of the transformed ranking vectors carry key information to detect the malicious nodes under Byzantine attacks. Based on these theoretical results, a highly efficient method called MANDERA is proposed to separate the malicious nodes from the benign ones by clustering all local nodes into two groups based on the ranking vectors. Figure 1 shows an illustrative motivation to our method. It demonstrates the behaviors of malicious and benign nodes under mean shift attacks. Obviously, the malicious and benign nodes are not distinguishable in the gradient space due to the challenges we mentioned above, while they are well separated in the ranking space.

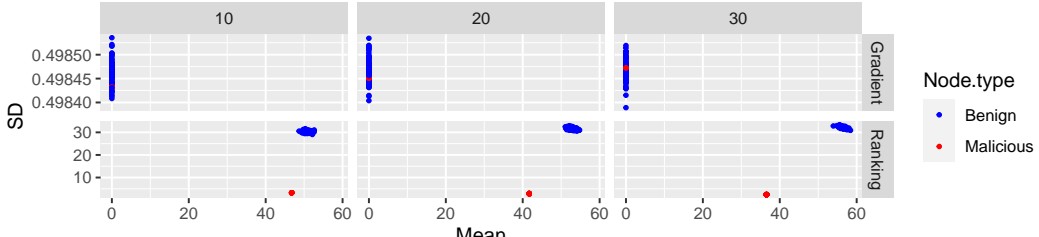

Figure 1: Patterns of nodes in gradient space and ranking space respectively under mean shift attacks. The columns of the figure represent the number of malicious nodes among 100 nodes: 10, 20 and 30.

The contributions of this work are as follows: **(1)** we propose the first algorithm leveraging the ranking space of model updates to detect malicious nodes (Figure 2); **(2)** we provide theoretical guarantee for the detection of malicious nodes based on the ranking space under Byzantine attacks; **(3)** our method does not assume knowledge on the number of malicious nodes, which is required in the learning process of most of the prior methods; **(4)** we experimentally demonstrate the effectiveness and robustness of our defense on Byzantine attacks, including Gaussian attack (GA), Sign Flipping attack (SF) and Zero Gradient attack (ZG) and Mean Shift attack (MF); **(5)** an experimental comparison between MANDERA and a collection of robust aggregation techniques is provided.

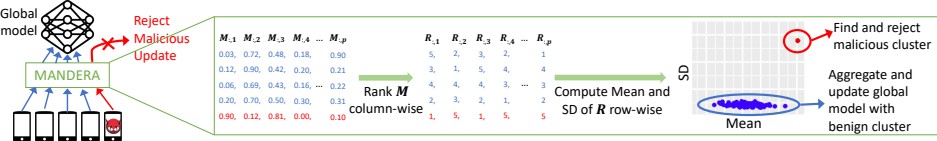

Figure 2: An overview of MANDERA.

**Related works.**   In the literature, there have been a collection of efforts along the research on defending Byzantine attacks. Blanchard et al. (2017) propose a defense referred to as Krum that treats local nodes whose update vector is too far away from the aggregated barycenter as malicious nodes and precludes them from the downstream aggregation. Guerraoui et al. (2018) propose Bulyan,

a process that performs aggregation on subsets of node updates (by iteratively leaving each node out) to find a set of nodes with the most aligned updates given an aggregation rule. Xie et al. (2019) compute a *Stochastic Descendant Score* (SDS) based on the estimated descendant of the loss function and the magnitude of the update submitted to the global node, and only include a predefined number of nodes with the highest SDS in the aggregation. On the other hand, Chen et al. (2021) propose a zero-knowledge approach to detect and remove malicious nodes by solving a weighted clustering problem. The resulting clusters update the model individually and accuracy against a validation set are checked. All nodes in a cluster with significant negative accuracy impact are rejected and removed from the aggregation step.

## 2 DEFENSE AGAINST BYZANTINE ATTACKS VIA RANKING

### 2.1 NOTATIONS

Suppose there are $n$ local nodes in the federated learning framework, where $n_1$ nodes are benign nodes whose indices are denoted by $\mathcal{I}_b$ and the other $n_0 = n - n_1$ nodes are malicious nodes whose indices are denoted by $\mathcal{I}_m$. The training model is denoted by $f(\boldsymbol{\theta}, \boldsymbol{D})$, where $\boldsymbol{\theta} \in \mathbb{R}^{p \times 1}$ is a $p$-dimensional parameter vector and $\boldsymbol{D}$ is a data matrix. Denote the message matrix received by the central server from all local nodes as $\boldsymbol{M} \in \mathbb{R}^{n \times p}$, where $\boldsymbol{M}_{i,:}$ denotes the message received from node $i$. For a benign node $i$, let $\boldsymbol{D}_i$ be the data matrix on it with $N_i$ as the sample size, we have $\boldsymbol{M}_{i,:} = \frac{\partial f(\boldsymbol{\theta}, \boldsymbol{D}_i)}{\partial \boldsymbol{\theta}} |_{\boldsymbol{\theta} = \boldsymbol{\theta}^*}$, where $\boldsymbol{\theta}^*$ is the parameter value from the global model. In the rest of the paper, we suppress $\frac{\partial f(\boldsymbol{\theta}, \boldsymbol{D}_i)}{\partial \boldsymbol{\theta}} |_{\boldsymbol{\theta} = \boldsymbol{\theta}^*}$ to $\frac{\partial f(\boldsymbol{\theta}, \boldsymbol{D}_i)}{\partial \boldsymbol{\theta}}$ to denote the gradient value for simplicity purpose. A malicious node $j \in \mathcal{I}_m$, however, tends to attack the learning system by manipulating $\boldsymbol{M}_{j,:}$ in some way. Hereinafter, we denote $N^* = \min(\{N_i\}_{i \in \mathcal{I}_b})$ to be the minimal sample size of the benign nodes.

Given a vector of real numbers $a \in \mathbb{R}^{n \times 1}$, define its ranking vector as $b = Rank(a) \in perm\{1, \cdots, n\}$, where the ranking operator $Rank$ maps the vector $a$ to an element in permutation space $perm\{1, \cdots, n\}$ which is the set of all the permutations of $\{1, \cdots, n\}$. For example, $Rank(1.1, -2, 3.2) = (2, 3, 1)$, it ranks the values from largest to smallest. We adopt average ranking, when there are ties. With the *Rank* operator, we can transfer the message matrix $\boldsymbol{M}$ to a ranking matrix $\boldsymbol{R}$ by replacing its column $\boldsymbol{M}_{:,j}$ by the corresponding ranking vector $\boldsymbol{R}_{:,j} = Rank(\boldsymbol{M}_{:,j})$. Further define

$$e_i \triangleq \frac{1}{p} \sum_{j=1}^{p} \boldsymbol{R}_{i,j} \qquad \text{and} \qquad v_i \triangleq \frac{1}{p} \sum_{j=1}^{p} (\boldsymbol{R}_{i,j} - e_i)^2$$

to be the mean and variance of $\boldsymbol{R}_{i,:}$, respectively. As it is shown in later subsections, we can judge whether node $i$ is a malicious node based on $(e_i, v_i)$ under various attack types. In the following, we will highlight the behavior of the benign nodes first, and then discuss the behavior of malicious nodes and their difference with the benign nodes under Byzantine attacks.

### 2.2 BEHAVIORS OF NODES UNDER BYZANTINE ATTACKS

Byzantine attacks aim to devastate the global model through manipulating the gradient values of some local nodes. For a general Byzantine attack, we assume that the gradient vectors of benign nodes and malicious nodes follow two different distributions $G$ and $F$. We would expect systematical difference on their behavior patterns in the ranking matrix $\boldsymbol{R}$, based on which malicious node detection can be achieved. Theorem 1 demonstrates the concrete behaviors of benign nodes and malicious nodes under general Byzantine attacks.

**Theorem 1** (Behavior under Byzantine attacks). *For a general Byzantine attack, assume that the gradient values from benign nodes and malicious nodes follow two distributions $G(\cdot)$ and $F(\cdot)$ respectively (both $G$ and $F$ are $p$-dimensional). We have*

$$\lim_{N^* \to \infty} \lim_{p \to \infty} e_i = \bar{\mu}_b \cdot \mathbb{I}(i \in \mathcal{I}_b) + \bar{\mu}_m \cdot \mathbb{I}(i \in \mathcal{I}_m) \ a.s.,$$

$$\lim_{N^* \to \infty} \lim_{p \to \infty} v_i = \bar{s}_b^2 \cdot \mathbb{I}(i \in \mathcal{I}_b) + \bar{s}_m^2 \cdot \mathbb{I}(i \in \mathcal{I}_m) \ a.s.,$$

*where $(\bar{\mu}_b, \bar{s}_b^2)$ and $(\bar{\mu}_m, \bar{s}_m^2)$ are highly non-linearly functions of $G(\cdot)$ and $F(\cdot)$ whose concrete form is detailed in the Appendix A, and "a.s." is the abbreviation of "almost surely".*

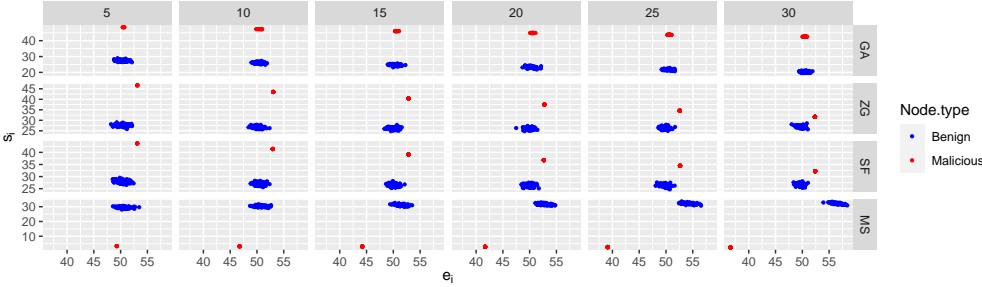

Figure 3: The scatter plots of $(e_i, s_i)$ for the 100 nodes under four types of attack as illustrative examples demonstrating ranking mean and standard deviation from the 1st epoch of training for the FASHION-MNIST dataset. Four attacks are Gaussian Attack (GA), Zero Gradient attack (ZG), Sign Flipping attack (SF) and Mean shift attack (MS).

The proof can be found in the Appendix A. If the attackers can access the exact distribution $G$, which is very rare, an obvious strategy to evade defense is to let $F = G$. For this case, the attack will have no impact to the global model. More often, the attackers have little information about distribution $G$. In this case, it is a rare event for the attackers to design a distribution $F$ satisfying $(\bar{\mu}_b, \bar{s}_b^2) = (\bar{\mu}_m, \bar{s}_m^2)$ for the malicious nodes to follow. In fact, most popular Byzantine attacks never try to make such an effort at all. Thus, the malicious nodes and the benign nodes are distinguishable with respect to their feature vectors $\{(e_i, v_i)\}_{1 \leq i \leq n}$, because $(e_i, v_i)$ reaches to different limits for begin and malicious nodes. Considering that the standard deviation $s_i = \sqrt{v_i}$ is typically of the similar scale of $e_i$, hereinafter we utilize $(e_i, s_i)$, instead of $(e_i, v_i)$, as the feature vector of node $i$ for malicious node detection.

Figure 3 illustrates the typical scatter plots of $(e_i, s_i)$ for benign and malicious nodes under four typical Byzantine attacks, i.e., GA, SN, ZG and MS. It can be observed that malicious nodes and benign nodes are all well separated in these scatter plots, indicating a proper clustering algorithm will distinguish these two groups. We note that both $s_i$ and $e_i$ are informative for malicious node detection, since in some cases (e.g., under Gaussian attacks) it is difficult to distinguish malicious nodes from the benign ones based on $e_i$ only.

## 2.3 Algorithm for Malicious node detection under Byzantine attacks

Theorem 1 implies that, under general Byzantine attacks, the feature vector $(e_i, s_i)$ of node $i$ converges to two different limits for benign and malicious nodes, respectively. Thus, for a real dataset where $N_i$'s and $p$ are all finite but reasonably large numbers, the scatter plot of $\{(e_i, s_i)\}_{1 \leq i \leq n}$ would demonstrate a clustering structure: one cluster for the benign nodes and the other cluster for the malicious nodes.

Based on this intuition, we propose *MAlicious Node DEtection via RAnking* (MANDERA) to detect the malicious nodes, whose workflow is detailed in Algorithm 1. MANDERA can be applied to either a single epoch or multiple epochs. For a single-epoch mode, the input data $M$ is the message matrix received from a single epoch. For multiple-epoch mode, the data $M$ is the column-concatenation of the message matrices from multiple epochs. By default, the experiments below all use single epoch to detect the malicious nodes.

---
**Algorithm 1** MANDERA

**Input:** The message matrix $M$.
1: Convert the message matrix $M$ to the ranking matrix $R$ by applying *Rank* operator.
2: Compute mean and standard deviation of rows in $R$, i.e., $\{(e_i, s_i)\}_{1 \leq i \leq n}$.
3: Run the clustering algorithm $K$-means to $\{(e_i, s_i)\}_{1 \leq i \leq n}$ with $K = 2$, and predict the set of benign nodes with the lager cluster denoted by $\hat{\mathcal{I}}_b$.

**Output:** The predicted benign node set $\hat{\mathcal{I}}_b$.

---

The predicted benign nodes $\hat{\mathcal{I}}_b$ obtained by MANDERA naturally leads to an aggregated message $\hat{m}_{b,:} = \frac{1}{\#(\hat{\mathcal{I}}_b)} \sum_{i \in \hat{\mathcal{I}}_b} M_{i,:}$. Theorem 2 shows that $\hat{\mathcal{I}}_b$ and $\hat{m}_b$ lead to consistent estimations of $\mathcal{I}_b$

and $\boldsymbol{m}_b = \frac{1}{n_1} \sum_{i \in \mathcal{I}_b} \boldsymbol{M}_{i,:}$ respectively, indicating that MANDERA enjoys *robustness guarantee* Steinhardt (2018) for Byzantine attacks.

**Theorem 2** (Robustness guarantee). *Under Byzantine attacks, we have:*

$$\lim_{N^* \to \infty} \lim_{p \to \infty} \mathbb{P}(\hat{\mathcal{I}}_b = \mathcal{I}_b) = 1, \quad and \quad \lim_{N^* \to \infty} \lim_{p \to \infty} \mathbb{E}||\hat{\boldsymbol{m}}_{b,:} - \boldsymbol{m}_{b,:}||_2 = 0. \tag{1}$$

The proof of Theorem 2 can be found in Appendix B. As $\mathbb{E}(\hat{\boldsymbol{m}}_{b,:}) = \boldsymbol{m}_{b,:}$, MANDERA obviously satisfies the $(\alpha, f)$-Byzantine Resilience condition, which is used in Blanchard et al. (2017) and Guerraoui et al. (2018) to measure the robustness of their estimators.

# 3 THEORETICAL ANALYSIS FOR SPECIFIC BYZANTINE ATTACKS

Theorem 1 provides us a general guidance about the behavior of nodes under Byzantine attacks. In this section, we examine the behavior for specific attacks, including Gaussian attacks, zero gradient attacks, sign flipping attacks and mean shift attack.

As the behavior of benign nodes does not depend on the type of Byzantine attack, we can study the statistical properties of $(e_i, v_i)$ for a benign node $i \in \mathcal{I}_b$ before the specification of a concrete attack type. For any benign node $i$, the message generated for $j^{th}$ parameter is $\boldsymbol{M}_{i,j} = \frac{1}{N_i} \sum_{l=1}^{N_i} \frac{\partial f(\boldsymbol{\theta}, \boldsymbol{D}_{i,l})}{\partial \boldsymbol{\theta}_j}$, where $\boldsymbol{D}_{i,l}$ denotes the $l^{th}$ sample on it. Throughout this paper, we assume that $\boldsymbol{D}_{i,l}$'s are independent and identically distributed (IID) samples drawn from a data distribution $\mathbb{D}$.

**Lemma 1.** *Under the IID data assumption, further denote* $\mu_j = \mathbb{E}\left(\frac{\partial f(\boldsymbol{\theta}, \boldsymbol{D}_{i,l})}{\partial \boldsymbol{\theta}_j}\right)$ *and* $\sigma_j^2 = \mathrm{Var}\left(\frac{\partial f(\boldsymbol{\theta}, \boldsymbol{D}_{i,l})}{\partial \boldsymbol{\theta}_j}\right) < \infty$, *with* $N_i$ *going to infinity, for* $\forall \ j \in \{1, \cdots, p\}$, *we have* $\boldsymbol{M}_{i,j} \to \mu_j$ *almost surely (a.s.) and* $\boldsymbol{M}_{i,j} \xrightarrow{d} \mathcal{N}\left(\mu_j, \sigma_j^2/N_i\right)$.

Lemma 1 can be proved by using the Kolmogorov's Strong Law of Large Numbers (KSLLN) and Central Limit Theorem. For the rest of this section, we will derive the detailed forms of $\bar{\mu}_b, \bar{\mu}_m, \bar{s}_b^2$ and $\bar{s}_m^2$, as defined in Theorem 1, under four specific Byzantine attacks.

## 3.1 GAUSSIAN ATTACK

**Definition 1** (Gaussian attack). In a Gaussian attack, the attacker generates malicious gradient values as follows: $\{\boldsymbol{M}_{i,:}\}_{i \in \mathcal{I}_m} \sim \mathcal{MVN}(\boldsymbol{m}_{b,:}, \Sigma)$, where $\boldsymbol{m}_{b,:} = \frac{1}{n_1} \sum_{i \in \mathcal{I}_b} \boldsymbol{M}_{i,:}$ is the mean vector of Gaussian distribution and $\Sigma$ is the covariance matrix determined by the attacker.

Considering that $\boldsymbol{M}_{i,j} \to \mu_j$ a.s. with $N_i$ going to infinity for all $i \in \mathcal{I}_b$ based on Definition 1, it is straightforward to see that $\lim_{N^* \to \infty} \boldsymbol{m}_{b,j} = \mu_j \ a.s.$, and the distribution of $\boldsymbol{M}_{i,j}$ for each $i \in \mathcal{I}_m$ converges to the Gaussian distribution centered at $\mu_j$. Based on this fact, the limiting behavior of the feature vector $(e_i, v_i)$ can be established for both benign and malicious nodes. Theorem 3 summarizes the results, with the proof detailed in Appendix C.

**Theorem 3** (Behavior under Gaussian attacks). *Assuming* $\{\boldsymbol{R}_{:,j}\}_{1 \le j \le p}$ *are independent of each other, under the Gaussian attack, the behaviors of benign and malicious nodes are as follows:*

$$\bar{\mu}_b = \bar{\mu}_m = \frac{n+1}{2}, \quad \bar{s}_b^2 = \frac{1}{p} \sum_{j=1}^p s_{b,j}^2, \quad \bar{s}_m^2 = \frac{1}{p} \sum_{j=1}^p s_{m,j}^2, \tag{2}$$

*where* $s_{b,j}^2$ *and* $s_{m,j}^2$ *are both complex functions of* $n_0, n_1, \sigma_j^2, \Sigma_{j,j}$ *and* $N^*$ *whose concrete form is detailed in the Appendix C.*

Considering that $\bar{s}_b^2 = \bar{s}_m^2$ if and only if $\Sigma_{j,j}$'s fall into a lower dimensional manifold whose measurement is zero under the Lebesgue measure, we have $P(\bar{s}_b^2 = \bar{s}_m^2) = 0$ if the attacker specifies the Gaussian variance $\Sigma_{j,j}$'s arbitrarily in the Gaussian attack. Thus, Theorem 3 in fact suggests that the benign nodes and the malicious nodes are different on the value of $v_i$, and therefore provides a guideline to detect the malicious nodes. Although the we do need $N^*$ and $p$ to go to infinity for getting the theoretical results in Theorem 3, in practice the malicious node detection algorithm based on the theorem typically works very well when $N^*$ and $p$ are reasonably large and $N_i$'s are not dramatically far away from each other.

The independent ranking assumption in Theorem 3, which assumes that $\{R_{:,j}\}_{1 \le j \le p}$ are independent of each other, may look restrictive. However, in fact it is a mild condition that can be easily satisfied in practice due to the following reasons. First, for a benign node $i \in \mathcal{I}_b$, $M_{i,j}$ and $M_{i,k}$ are often nearly independent, as the correlation between two model parameters $\boldsymbol{\theta}_j$ and $\boldsymbol{\theta}_k$ is often very weak in a large

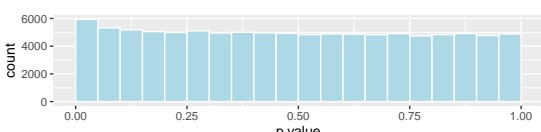

Figure 4: Independence test for 100,000 column pairs randomly chosen from message matrix $M$ generated from FASHION-MNIST data.

deep neural network with a huge number of parameters. To verify the statement, we implemented independence tests for 100,000 column pairs randomly chosen from the message matrix $M$ generated from the FASHION-MNIST data. Distribution of the p-values of these tests are demonstrated in Figure 4 via a histogram, which is very close to a uniform distribution, indicating that $M_{i,j}$ and $M_{i,k}$ are indeed nearly independent in practice. Second, even some $M_{:,j}$ and $M_{:,k}$ shows strong correlation, magnitude of the correlation would be reduced greatly during the transformation from $M$ to $R$, as the final ranking $R_{i,j}$ also depends on many other factors. Actually, the independent ranking assumption could be relaxed to be uncorrelated ranking assumption which assumes the rankings are uncorrelated with each other. Adopting the weaker assumption will result in a change of convergence type of our theorems from the "almost surely convergence" to "convergence in probability".

### 3.2 SIGN FLIPPING ATTACK

**Definition 2** (Sign flipping attack). Sign flipping attack aims to generate the gradient values of malicious nodes by flipping the sign of the average of all the benign nodes' gradient at each epoch, i.e., specifying $M_{i,:} = -r m_{b,:}$ for any $i \in \mathcal{I}_m$, where $r > 0$, $m_b = \frac{1}{n_1} \sum_{k \in \mathcal{I}_b} M_{k,:}$.

Based on the above definition, the update message of a malicious node $i$ under the sign flipping attack is $M_{i,:} = -r m_{b,:} = -\frac{r}{n_1} \sum_{k \in \mathcal{I}_b} M_{k,:}$. The theorem 4 summarizes the behavior of malicious nodes and benign nodes respectively, with the detailed proof provided in Appendix D.

**Theorem 4** (Behavior under sign flipping attacks). *With the same assumption as posed in Theorem 3, under the sign flipping attack, the behaviors of benign and malicious nodes are as follows:*

$$\bar{\mu}_b = \frac{n+n_0+1}{2} - n_0\rho, \quad \bar{\mu}_m = n_1\rho + \frac{n_0+1}{2}, \tag{3}$$

$$\bar{s}_b^2 = \rho S_{[1,n_1]}^2 + (1-\rho)S_{[n_0+1,n]}^2 - (\bar{\mu}_b)^2, \quad \bar{s}_m^2 = \rho S_{[n_1+1,n]}^2 + (1-\rho)S_{[1,n_0]}^2 - (\bar{\mu}_m)^2, \tag{4}$$

*where $\rho = \lim_{p \to \infty} \frac{\sum_{j=1}^p \mathbb{I}(\mu_j > 0)}{p}$, $S_{[a,b]}^2 = \frac{1}{b-a+1} \sum_{k=a}^b k^2$, $\bar{s}_m^2$ and $\bar{s}_b^2$ are both quadratic functions of $\rho$ whose concrete form also depends on $n_0$ and $n_1$.*

Considering that $\bar{\mu}_b = \bar{\mu}_m$ if and only if $\rho = \frac{1}{2}$, and $\bar{s}_b^2 = \bar{s}_m^2$ if and only if $\rho$ is the solution of a quadratic function, the probability of $(\bar{\mu}_b, \bar{s}_b^2) = (\bar{\mu}_m, \bar{s}_m^2)$ is zero as $p \to \infty$. Such a phenomenon suggests that we can detect the malicious nodes based on the moments $(e_i, v_i)$ to defense the sign flipping attack as well. Noticeably, we note that the limit behavior of $e_i$ and $v_i$ does not dependent on the specification of $r$, which defines the sign flipping attack. Although such a fact looks a bit abnormal at the first glance, it is totally understandable once we realize that with the variance of $M_{i,j}$ shrinks to zero with $N_i$ goes to infinity for each benign node $i$, any different between $\mu_j$ and $\mu_j(r)$ would result in the same ranking vector $R_{:,j}$ in the ranking space.

### 3.3 ZERO GRADIENT ATTACK

**Definition 3** (Zero gradient attack). Zero gradient attack aims to make the aggregated message to be zero, i.e., $\sum_{i=1}^n M_{i,:} = 0$, at each epoch, by specifying $M_{i,:} = -\frac{n_1}{n_0} m_{b,:}$ for all $i \in \mathcal{I}_m$.

Apparently, the zero gradient attack defined above is a special case of sign flipping attack by specifying $r = \frac{n_1}{n_0}$. The conclusions of Theorem 4 keep unchanged for different specifications of $r$. Therefore, the behavior follows the same limiting behaviors as described in Theorem 4.

### 3.4 MEAN SHIFT ATTACK

**Definition 4** (Mean shift attack). Mean shift attack (Baruch et al., 2019) manipulates the updates of the malicious nodes in the following fashion, $\boldsymbol{m}_{i,j} = \mu_j - z \cdot \sigma_j$ for $i \in \mathcal{I}_m$ and $1 \le j \le p$, where $\mu_j = \frac{1}{n_1} \sum_{i \in \mathcal{I}_b} \boldsymbol{M}_{i,j}, \sigma_j = \sqrt{\frac{1}{n_1} \sum_{i \in \mathcal{I}_b} (\boldsymbol{M}_{i,j} - \mu_j)^2}$ and $z = \arg\max_t \phi(t) < \frac{n-2}{2(n-n_0)}$.

Mean shift attacks aim to generate malicious gradients which are not well separated, but of different distributions, from the benign nodes. Theorem 5 details the behavior of malicious nodes and benign nodes under mean shift attacks. The proof can be found in Appendix E

**Theorem 5.** *With the same assumption as posed in Theorem 3 and additionally $n$ is relatively large, under the mean shift attack, the behaviors of benign and malicious nodes are as follows:*

$$\bar{\mu}_b = \frac{n+1}{2} + \frac{n_0}{n_1}(n_1 - \alpha), \quad \bar{\mu}_m = \alpha + \frac{n_0+1}{2}, \tag{5}$$

$$\bar{s}_b^2 = \frac{1}{n_1}\left(\tau(n) + \tau(\alpha) - \tau(\alpha + 1 + n_0)\right) - \bar{\mu}_b^2, \quad \bar{s}_m^2 = 0, \tag{6}$$

*where $\lfloor \cdot \rfloor$ denotes the floor function, $\alpha = \lfloor n_1 \Phi(z) \rfloor$, $\Phi(z)$ is the cumulative density function of the standard normal distribution and $\tau(\cdot)$ is the function of 'sum of squares', i.e., $\tau(n) = \sum_{k=1}^n k^2$.*

## 4 EXPERIMENTS

In these experiments we extend the data poisoning experimental framework of Tolpegin et al. (2020); Wu et al. (2020a), integrating Byzantine attack implementations released by Wu et al. (2020b) and the mean shift attack Baruch et al. (2019). The mean shift attack was designed to poison gradients by adding 'a little' amount of noise, and shown to be effective in defeating Krum (Blanchard et al., 2017) and Bulyan (Guerraoui et al., 2018) defenses. The mean shift attack is defined in Definition 4. In our experiments, we set $\Sigma = 30\boldsymbol{I}$ for the Gaussian attack and $r = 3$ for the sign flipping attack, where $\boldsymbol{I}$ is the identity matrix. For all experiments we fix $n = 100$ participating nodes, of which a variable number of nodes are poisoned $|n_0| \in \{5, 10, 15, 20, 25, 30\}$. The training process is run until 25 epochs have elapsed. We have described the structure of these networks in Appendix F.

### 4.1 DEFENSE BY MANDERA FOR IID SETTINGS

We evaluate the efficacy in detecting malicious nodes within the federated learning framework with the use of three IID datasets. The first is the FASHION-MNIST dataset Xiao et al. (2017), a dataset of 60,000 and 10,000 training and testing samples respectively divided into 10 classes of apparel. The second is CIFAR-10 Krizhevsky et al. (2009), a dataset of 60,000 small object images also containing 10 object classes. The third is the MNIST Deng (2012) dataset. The MNIST dataset is a dataset of 60,000 and 10,000 training and testing samples respectively divided into 10 classes of handwritten digits from multiple authors.

We test the performance of MANDERA on the update gradients of a model under attacks. In this section, MANDERA acts as an observer without intervening in the learning process to identify malicious nodes with a set of gradients from a single epoch. Each configuration of 25 training epochs, with a given number of malicious nodes was repeated 20 times. Figure 5 demonstrates the classification performance (Metrics defined in Appendix G) of MANDERA with different settings of participating malicious nodes and the four poisoning attacks, i.e., GA, ZG, SF and MS.

While we have formally demonstrated the efficacy of MANDERA in accurately detecting potentially malicious nodes participating in the federated learning process. In practice, to leverage an unsupervised K-means clustering algorithm, we must also identify the correct group of nodes as the malicious group. Our strategy is to identify the group with the most exact gradients, or otherwise the smaller group (we regard a system with over 50% of their nodes compromised as having larger issues than just poisoning attacks). [1] We also test other clustering algorithms, such as hierarchical clustering and Gaussian mixture models Fraley and Raftery (2002). It turns out that the performance of MANDERA is quite robust with different choices of clustering methods. Detailed results can

---

[1]More informed approaches to selecting the malicious cluster can be tested in future work. E.g. Figure 3 displays less variation of ranking variance in malicious cluster compared to benign nodes. This could robust selection of the malicious group, and enabling selection of malicious groups larger than 50%.

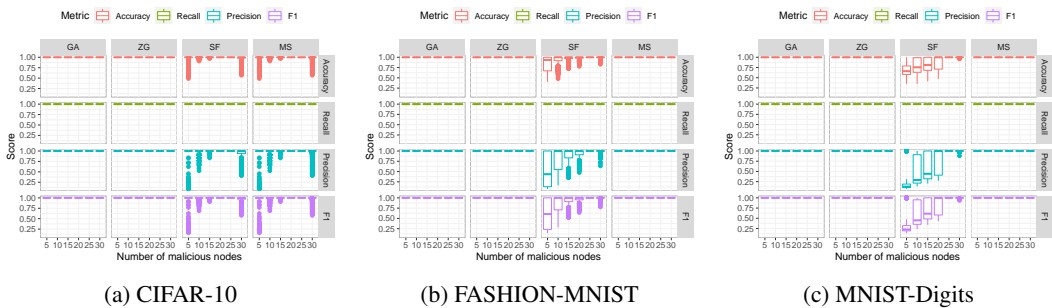

(a) CIFAR-10  (b) FASHION-MNIST  (c) MNIST-Digits

Figure 5: Classification performance of our proposed approach MANDERA under four types of attack for three IID settings.

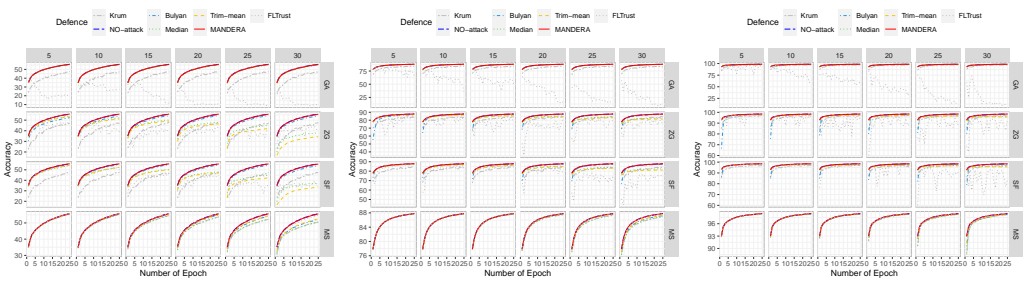

(a) CIFAR-10 accuracy  (b) FASHION-MNIST accuracy  (c) MNIST-Digits accuracy

Figure 6: Model Accuracy at each epoch of training, each line of the curve represents a different defense against the Byzantine attacks.

be found in Appendix H. From Figure 5, it is immediately evident that the recall of the malicious nodes for the Byzantine attacks is exceptional. However, occasionally benign nodes have also been misclassified as malicious under SF attacks. On all attacks, in the presence of more malicious nodes, the recall of malicious nodes trends down.

We encapsulate MANDERA into a module prior to the the aggregation step, MANDERA has the sole objective of identifying malicious nodes, and excluding their updates from the global aggregation step. Each configuration of 25 training epochs, a given poisoning attack, defense method, and a given number of malicious nodes was repeated 10 times. We compare MANDERA against 5 other robust aggregation defense methods, Krum Blanchard et al. (2017), Bulyan Guerraoui et al. (2018), Trimmed Mean Yin et al. (2018), Median Yin et al. (2018) and FLTrust Cao et al. (2020). Of which the first 2 requires an assumed number of malicious nodes, and the latter 3 only aggregate robustly.

From Figure 6, it is observed that MANDERA outperforms the competing methods uniformly, approaching to the performance of a model not under attack. The corresponding model loss can be found in Appendix I. Interestingly, FLTrust as a standalone defense is weak in protecting against the most extreme Byzantine attacks. However, we highlight that FLtrust is a robust aggregation method against targeted attacks that may thwart defences like Krum, Trimmed mean. We see FLTrust as a complementary defence that relies on a base method of defence against Byzantine attacks, but expands the protection coverage of the FL system against adaptive attacks.

## 4.2 DEFENSE BY MANDERA FOR NON-IID SETTINGS

In this section, we evaluate applicability of MANDERA when applied in a non-IID setting in Federated learning to validate its effectiveness. The batch size present through the existing evaluations of Section 4.1 is 10, this low setting practically yields gradient values at each local worker node as if they were derived from non-IID samples. This is a strong indicator that MANDERA could be effective for non-IID settings. We reinforce MANDERA's applicability in the non-IID setting by repeating the experiment on QMNIST Yadav and Bottou (2019), a dataset that is per-sample equivalent to MNIST Deng (2012). QMNIST, however, additionally provides us with writer identification information. This identity is leverages to ensure that each local node only trains on digits written by a

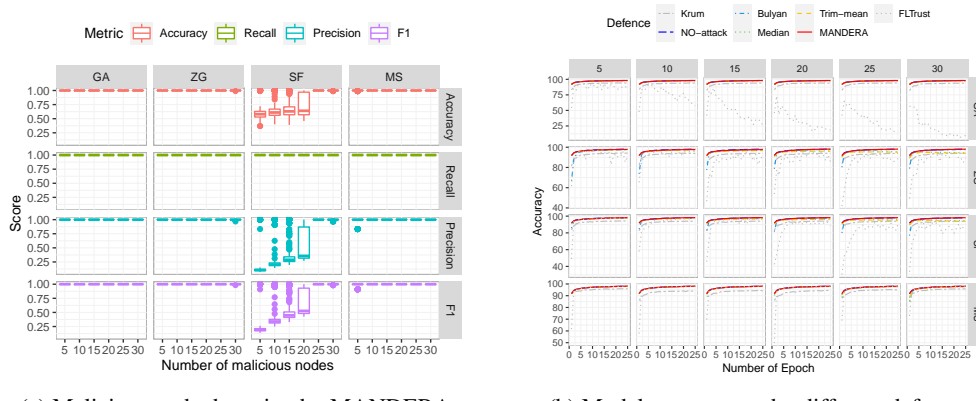

(a) Malicious node detection by MANDERA          (b) Model accuracy under different defenses

Figure 7: Results for a Non-IID setting: QMNIST dataset under four different Byzantine attacks.

set of unique users not seen by other workers. Such a setting is widely recognized as non-IID setting in the community (Kairouz et al., 2021). For 100 nodes, this works out to be approximately 5 writers in each node. All other experimental configurations remain the same as Section 4.1.

Figure 7a demonstrates the effectiveness of MANDERA in malicious node detection for the non-IID setting. It performs near perfectly under Gaussian attacks, zero gradient attacks and Mean shift attacks. On the other hand, when the number of malicious nodes is less than 25, MANDERA mis-classifies some benign nodes as malicious under sign-flipping attacks. However, it is still capable to detect all the malicious nodes. This behavior is consistent with the behavior observed for MNIST in Figure 5c. Figure 7b shows the global model training accuracy with different defense strategies for a non-IID setting. The figure indicates that MANDERA outperforms the other defending strategies and achieves about the best performance where no attack is conduct to the global model. The model loss can be found in Appendix J. MANDERA enjoys super-fast computation. We have listed the computational times of state-of-art methods in Appendix K.

## 5    DISCUSSION AND CONCLUSION

Theorem 1 indicates that Byzantine attacks can only evade MANDERA when the attackers know the distribution of benign nodes and at the same time huge computational resources are required. This makes MANDERA a strategy which is challenging for attackers to evade.

We acknowledge FL framework may learn the global model only using subset of nodes at each round. In these settings MANDERA would still function, as we would rank and cluster on the parameters of the participating nodes, without assuming any number of poisoned nodes. In Algorithm 1, performance could be improved by incorporating higher order moments. MANDERA is unable to function when gradients are securely aggregated in its current form. However, malicious nodes can be identified and excluded from the secure aggregation step, while still protecting the privacy of participating nodes by performing MANDERA through secure ranking Zhang et al. (2013); Lin and Tzeng (2005) (recall that MANDERA only requires the ranking matrix to detect poisoned nodes).

In conclusion, we proposed a novel way to tackle the challenges for malicious node detection when using the gradient values. Our method transfers the gradient values to a ranking space. We have provided theoretical guarantees and experimentally shown efficacy in MANDERA for the detection of malicious nodes performing poisoning attacks against federated learning. Our proposed method MANDERA, is able to achieve excellent detection accuracy and maintain a higher model accuracy than other seminal.

## Ethics Statement

The core objective of our research is to provide an additional means of defense against poisoning nodes that target Federated Learning. To test our defense we have implemented different attacks against the Federated Learning framework. Attackers may adopt our defense strategy to design new poisoning attacks. Fortunately, these poisoning attacks can not be leveraged to leak private information from Federated learning models, instead only impact its performance.

## Reproducibility Statement

To ensure reproducible research, we have supplemented our proposal for MANDERA, by supplying both R and Python implementations of MANDERA used in this paper, uploaded with the remainder of the experiment code. The four datasets featured in this paper are CIFAR-10 (Krizhevsky et al., 2009), Fasion-MNIST (Xiao et al., 2017), MNIST (Deng, 2012) and QMNIST Yadav and Bottou (2019); we have used each of these dataset unaltered from their respective sources. We have stated the assumptions in our theorems and their proofs can be found in the Appendix. But to explain our assumptions in simple terms, (1) The data samples on each local node are independently drawn from the same distribution. (2) The gradient value for each parameter is independent to each other.

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

## APPENDIX

## A   PROOF OF THEOREM 1

*Proof.* Let $F_j(x)$ and $G_j(x)$ be the cumulative distribution functions of $F_j(\cdot)$ and $G_j(\cdot)$, $f_j(x)$ and $g_j(x)$ be the corresponding density functions, and $r_j(x) = n_1 - n_1 G_j(x) + n_0 - n_0 F_j(x) + 1$ be the expected ranking of value $x$ among all entries in the $j^{th}$ column of the gradient value matrix.

Further define

$$E_{bj} = \int_{-\infty}^{\infty} r_j(x) g_j(x) dx, \; V_{bj} = \int_{-\infty}^{\infty} (r_j(x) - E_{bj})^2 g_j(x) dx,$$

$$E_{mj} = \int_{-\infty}^{\infty} r_j(x) f_j(x) dx, \; V_{mj} = \int_{-\infty}^{\infty} (r_j(x) - E_{mj})^2 f_j(x) dx.$$

It can be shown for any $1 \le j \le p$ that

$$
\begin{aligned}
E_{ij} &= \mathbb{E}(\mathbf{R}_{i,j}) = E_{bj} \cdot \mathbb{I}(i \in \mathcal{I}_b) + E_{mj} \cdot \mathbb{I}(i \in \mathcal{I}_m), \\
V_{ij} &= \mathbb{V}(\mathbf{R}_{i,j}) = V_{bj} \cdot \mathbb{I}(i \in \mathcal{I}_b) + V_{mj} \cdot \mathbb{I}(i \in \mathcal{I}_m).
\end{aligned}
$$

Thus, we would have according to Kolmogorov's strong law of large numbers (KSLLN) that

$$
\begin{aligned}
\lim_{N^* \to \infty} \lim_{p \to \infty} e_i &= \bar{\mu}_b \cdot \mathbb{I}(i \in \mathcal{I}_b) + \bar{\mu}_m \cdot \mathbb{I}(i \in \mathcal{I}_m) \; a.s., \\
\lim_{N^* \to \infty} \lim_{p \to \infty} v_i &= \bar{s}_b^2 \cdot \mathbb{I}(i \in \mathcal{I}_b) + \bar{s}_m^2 \cdot \mathbb{I}(i \in \mathcal{I}_m) \; a.s.,
\end{aligned}
$$

where the moments $(\bar{\mu}_b, \bar{s}_b^2)$ and $(\bar{\mu}_m, \bar{s}_m^2)$ are deterministic functions of $(E_{bj}, V_{bj})$ and $(E_{mj}, V_{mj})$ of the following form:

$$\bar{\mu}_b = \lim_{p \to \infty} \frac{1}{p} \sum_{j=1}^{p} E_{bj}, \qquad \bar{\mu}_m = \lim_{p \to \infty} \frac{1}{p} \sum_{j=1}^{p} E_{mj},$$

$$\bar{s}_b^2 = \lim_{p \to \infty} \frac{1}{p} \sum_{j=1}^{p} V_{bj}, \qquad \bar{s}_m^2 = \lim_{p \to \infty} \frac{1}{p} \sum_{j=1}^{p} V_{mj}.$$

It completes the proof. $\qquad \square$

## B  PROOF OF THEOREM 2

*Proof.* According to Theorem 1, when both $N^*$ and $p$ are large enough, with probability 1 there exist $(e_b, v_b)$, $(e_m, v_m)$ and $\delta > 0$ such that $||(e_b, v_b) - (e_m, v_m)||_2 > \delta$, and

$$||(e_i, v_i) - (e_b, v_b)||_2 \le \frac{\delta}{2} \; \text{for} \; \forall \, i \in \mathcal{I}_b \quad \text{and} \quad ||(e_i, v_i) - (e_m, v_m)||_2 \le \frac{\delta}{2} \; \text{for} \; \forall \, i \in \mathcal{I}_m.$$

Therefore, with a reasonable clustering algorithm such as $K$-mean with $K = 2$, we would expect $\hat{\mathcal{I}}_b = \mathcal{I}_b$ with probability 1.

Because we can always find a $\Delta > 0$ such that $||\mathbf{M}_{i,:} - \mathbf{M}_{j,:}||_2 \le \Delta$ for any node pair $(i, j)$ in a fixed dataset with a finite number of nodes, and $\hat{\mathbf{m}}_{b,:} = \mathbf{m}_{b,:}$ when $\hat{\mathcal{I}}_b = \mathcal{I}_b$, we have

$$\mathbb{E}||\hat{\mathbf{m}}_{b,:} - \mathbf{m}_{b,:}||_2 \le \Delta \cdot \mathbb{P}(\hat{\mathcal{I}}_b \ne \mathcal{I}_b),$$

and thus

$$\lim_{N^* \to \infty} \lim_{p \to \infty} \mathbb{E}||\hat{\mathbf{m}}_{b,:} - \mathbf{m}_{b,:}||_2 = 0.$$

It completes the proof. $\qquad \square$

## C  PROOF OF THEOREM 3

*Proof.* According to Theorem 1, we only need to compute $\bar{\mu}_b, \bar{\mu}_m, \bar{s}_b^2$ and $\bar{s}_m^2$ under the Gaussian attacks.

Because $\mathbf{M}_{i,j} \xrightarrow{d} \mathcal{N}(\mu_j, \Sigma_{j,j})$ for $\forall \, i \in \mathcal{I}_m$ and $\mathbf{M}_{i,j} \xrightarrow{d} \mathcal{N}(\mu_j, \sigma_j^2 / N_i)$ for $\forall \, i \in \mathcal{I}_b$ when $N^* \to \infty$, it is straightforward to see due to the symmetry of Gaussian distribution that

$$\lim_{N^* \to \infty} E_{bj} = \lim_{N^* \to \infty} E_{mj} = \lim_{N^* \to \infty} \mathbb{E}(\mathbf{R}_{i,j}) = \frac{n+1}{2}, \; 1 \le i \le n, \; 1 \le j \le p. \qquad (7)$$

Therefore, we have

$$\bar{\mu}_b = \lim_{N^* \to \infty} \lim_{p \to \infty} \frac{1}{p} \sum_{j=1}^{p} E_{bj} = \frac{n+1}{2},$$

$$\bar{\mu}_m = \lim_{N^* \to \infty} \lim_{p \to \infty} \frac{1}{p} \sum_{j=1}^{p} E_{mj} = \frac{n+1}{2}.$$

Moreover, assuming that the sample sizes of different benign nodes approach to each other with $N^*$ going to infinity, i.e.,

$$\lim_{N^* \to \infty} \frac{1}{N^*} \max_{i,k \in \mathcal{I}_b} |N_i - N_k| = 0, \tag{8}$$

for each parameter dimension $j$, $\{\boldsymbol{M}_{i,j}\}_{i \in \mathcal{I}_b}$ would converge to the same Gaussian distribution $\mathcal{N}(\mu_j, \sigma_j^2/N^*)$ with the increase of $N^*$. Thus, due to the exchangeability of $\{\boldsymbol{M}_{i,j}\}_{i \in \mathcal{I}_b}$ and $\{\boldsymbol{M}_{i,j}\}_{i \in \mathcal{I}_m}$, it is easy to see that that

$$\lim_{N^* \to \infty} V_{bj} = s_{b,j}^2, \quad \lim_{N^* \to \infty} V_{mj} = s_{m,j}^2, \tag{9}$$

where $s_{b,j}^2$ and $s_{m,j}^2$ are both complex functions of $n_0$, $n_1$, $\sigma_j^2$, $\Sigma_{j,j}$ and $N^*$, and $s_{b,j}^2 = s_{m,j}^2$ if and only if $\sigma_j^2/N^* = \Sigma_{j,j}$. According to Theorem 1, $\bar{s}_b^2 = \lim_{p \to \infty} \frac{1}{p} \sum_{j=1}^{p} V_{bj} = \lim_{N^* \to \infty} \frac{1}{p} \sum_{j=1}^{p} s_{b,j}^2$ and $\bar{s}_m^2 = \lim_{p \to \infty} \frac{1}{p} \sum_{j=1}^{p} V_{mj} = \lim_{p \to \infty} \frac{1}{p} \sum_{j=1}^{p} s_{m,j}^2$. The proof is complete. □

## D    PROOF OF THEOREM 4

*Proof.* According to Theorem 1, we only need to compute $\bar{\mu}_b, \bar{\mu}_m, \bar{s}_b^2$ and $\bar{s}_m^2$ under the sign flipping attacks.

**Lemma 2.** *Under the sign flipping attack, for each malicious node $i \in \mathcal{I}_m$ and any parameter dimension $j$, we have $\boldsymbol{M}_{i,j} = -\frac{r}{n_1} \sum_{k \in \mathcal{I}_b} \boldsymbol{M}_{k,j}$ is a deterministic function of $\{\boldsymbol{M}_{k,j}\}_{k \in \mathcal{I}_b}$, whose limiting distribution when $N^*$ goes to infinity is*

$$\boldsymbol{M}_{i,j} \xrightarrow{d} \mathcal{N}(\mu_j(r), \sigma_j^2(r)), \ 1 \le j \le p, \tag{10}$$

*where $\mu_j(r) = -r\mu_j$, $\sigma_j^2(r) = \frac{r^2 \cdot \sigma_j^2}{n_1 \cdot \bar{N}_b}$, and $\bar{N}_b = \frac{n_1}{\sum_{k \in \mathcal{I}_b} \frac{1}{N_k}}$ is the harmonic mean of $\{N_k\}_{k \in \mathcal{I}_b}$.*

Lemma 1 and Lemma 2 tell us that for each parameter dimension $j$, the distribution of $\{\boldsymbol{M}_{i,j}\}_{i=1}^{n}$ is a mixture of Gaussian components $\{\mathcal{N}(\mu_j, \sigma_j^2/N_i)\}_{i \in \mathcal{I}_b}$ centered at $\mu_j$ plus a point mass located at $\mu_j(r) = -r\mu_j$. If $N_i$'s are reasonably large, variances $\sigma_j^2/N_i$'s would be very close to zero, and the probability mass of the mixture distribution would concentrate to two local centers $\mu_j$ and $\mu_j(r) = -r\mu_j$, one for the benign nodes and the other one for the malicious nodes.

Under the sign flipping attack, because $\boldsymbol{M}_{i,j} \xrightarrow{d} \mathcal{N}(\mu_j(r), \sigma_j^2(r))$ for $\forall i \in \mathcal{I}_m$ and $\boldsymbol{M}_{i,j} \xrightarrow{d} \mathcal{N}(\mu_j, \sigma_j^2/N_i)$ for $\forall i \in \mathcal{I}_b$ when $N^* \to \infty$, and

$$\lim_{N^* \to \infty} (\sigma_j^2/N_i) = \lim_{N^* \to \infty} \sigma_j^2(r) = 0.$$

It is straightforward to see that

$$\lim_{N^* \to \infty} P(\boldsymbol{M}_{i,j} > \boldsymbol{M}_{k,j}) = \mathbb{I}(\mu_j > 0), \ \forall i \in \mathcal{I}_b, \forall k \in \mathcal{I}_m,$$

which further indicates that

$$\begin{aligned}
\lim_{N^* \to \infty} E_{bj} &= \lim_{N^* \to \infty} \mathbb{E}(\boldsymbol{R}_{i,j}) = \frac{n_1 + 1}{2}, \ if \ \mu_j > 0, \\
\lim_{N^* \to \infty} E_{mj} &= \lim_{N^* \to \infty} \mathbb{E}(\boldsymbol{R}_{i,j}) = \frac{n + n_1 + 1}{2}, \ if \ \mu_j > 0, \\
\lim_{N^* \to \infty} E_{bj} &= \lim_{N^* \to \infty} \mathbb{E}(\boldsymbol{R}_{i,j}) = \frac{n + n_0 + 1}{2} \ if \ \mu_j < 0 \\
\lim_{N^* \to \infty} E_{mj} &= \lim_{N^* \to \infty} \mathbb{E}(\boldsymbol{R}_{i,j}) = \frac{n_0 + 1}{2} \ if \ \mu_j < 0,
\end{aligned} \tag{11}$$

$$\lim_{N^* \to \infty} \mathbb{E}(\boldsymbol{R}_{i,j}^2) = S_{[1,n_1]}^2 \cdot \mathbb{I}(i \in \mathcal{I}_b) + S_{[n_1+1,n]}^2 \cdot \mathbb{I}(i \in \mathcal{I}_m) \; if \; \mu_j > 0,$$
$$\lim_{N^* \to \infty} \mathbb{E}(\boldsymbol{R}_{i,j}^2) = S_{[1,n_0]}^2 \cdot \mathbb{I}(i \in \mathcal{I}_m) + S_{[n_0+1,n]}^2 \cdot \mathbb{I}(i \in \mathcal{I}_b) \; if \; \mu_j < 0, \tag{12}$$

where $S_{[a,b]}^2 = \frac{1}{b-a+1} \sum_{k=a}^{b} k^2$.

Therefore, we have

$$\begin{cases} \bar{\mu}_m = \lim_{N^* \to \infty} \lim_{p \to \infty} \frac{1}{p} \sum_{j=1}^{p} E_{bj} = \rho \cdot \frac{n+n_1+1}{2} + (1-\rho) \cdot \frac{n_0+1}{2}, \\ \bar{\mu}_b = \lim_{N^* \to \infty} \lim_{p \to \infty} \frac{1}{p} \sum_{j=1}^{p} E_{mj} \rho \cdot \frac{n_1+1}{2} + (1-\rho) \cdot \frac{n+n_0+1}{2}, \end{cases}$$

where $\rho = \lim_{p \to \infty} \frac{\sum_{j=1}^{p} \mathbb{I}(\mu_j > 0)}{p}$.

Define $\bar{\mu}_i = \bar{\mu}_m \cdot \mathbb{I}(i \in \mathcal{I}_m) + \bar{\mu}_b \cdot \mathbb{I}(i \in \mathcal{I}_b)$. Considering that

$$\lim_{N^* \to \infty} \lim_{p \to \infty} \frac{1}{p} \sum_{j=1}^{p} V_{ij}$$

$$= \lim_{N^* \to \infty} \lim_{p \to \infty} \frac{1}{p} \sum_{j=1}^{p} \mathbb{E}(\boldsymbol{R}_{i,j} - \bar{\mu}_i)^2$$

$$= \lim_{p \to \infty} \lim_{N^* \to \infty} \frac{1}{p} \sum_{j=1}^{p} \left( \mathbb{E}(\boldsymbol{R}_{i,j}^2) - 2\bar{\mu}_i \mathbb{E}(\boldsymbol{R}_{i,j}) + (\bar{\mu}_i)^2 \right)$$

$$= \left[ \bar{\tau}_m - (\bar{\mu}_m)^2 \right] \cdot \mathbb{I}(i \in \mathcal{I}_m) + \left[ \bar{\tau}_b - (\bar{\mu}_b)^2 \right] \cdot \mathbb{I}(i \in \mathcal{I}_b),$$

where

$$\bar{\tau}_b = \rho \cdot S_{[1,n_1]}^2 + (1-\rho) \cdot S_{[n_0+1,n]}^2,$$
$$\bar{\tau}_m = \rho \cdot S_{[n_1+1,n]}^2 + (1-\rho) \cdot S_{[1,n_0]}^2.$$

According to Theorem 1,

$$\bar{s}_b^2 = \lim_{p \to \infty} \lim_{N^* \to \infty} \frac{1}{p} \sum_{j=1}^{p} V_{bj} = \bar{\tau}_b - (\bar{\mu}_b)^2,$$

$$\bar{s}_m^2 = \lim_{p \to \infty} \lim_{N^* \to \infty} \frac{1}{p} \sum_{j=1}^{p} V_{mj} = \bar{\tau}_m - (\bar{\mu}_m)^2.$$

It completes the proof. $\qquad\square$

## E   PROOF OF THEOREM 5

*Proof.* According to Theorem 1, we only need to compute $\bar{\mu}_b, \bar{\mu}_m, \bar{s}_b^2$ and $\bar{s}_m^2$ under the mean shift attacks.

Under the mean shift attack, all the malicious gradient will be inserted at a position which is dependent on $z$. More specifically, for a relatively large $n$, the samples from benign nodes are normally distributed. Therefore, on average, with proportion $\Phi(z)$ of the benign nodes having higher values of gradient than the malicious nodes.

First of all, we derive the property in term of the first moment. Denote $\alpha = \lfloor n_1 \Phi(z) \rfloor$. For a benign node, we have

$$\lim_{N^* \to \infty} \lim_{n \to \infty} E_{bj} = \lim_{N^* \to \infty} \lim_{n \to \infty} \mathbb{E}(\boldsymbol{R}_{i,j}) = \frac{1}{n_1} \left( \sum_{k=1}^{\alpha} k + \sum_{s=n_0+1+\alpha}^{n} s \right) = \frac{n+1}{2} + \frac{n_0}{n_1}(n_1 - \alpha).$$

For a malicious node, we have

$$\lim_{N^* \to \infty} \lim_{n \to \infty} E_{mj} = \lim_{N^* \to \infty} \lim_{n \to \infty} \mathbb{E}(\boldsymbol{R}_{i,j}) = \frac{\alpha + 1 + \alpha + n_0}{2} = \alpha + \frac{1 + n_0}{2}.$$

Therefore, according to Theorem 1,

$$
\begin{aligned}
\bar{\mu}_b &= \lim_{N^*\to\infty} \lim_{n\to\infty} \lim_{p\to\infty} \frac{1}{p} \sum_{j=1}^{p} E_{bj} = \frac{n+1}{2} + \frac{n_0}{n_1}(n_1 - \alpha), \\
\bar{\mu}_m &= \lim_{N^*\to\infty} \lim_{n\to\infty} \lim_{p\to\infty} \frac{1}{p} \sum_{j=1}^{p} E_{mj} = \alpha + \frac{1+n_0}{2}.
\end{aligned}
$$

Now, we derive the property in term of the second moment. For a benign node, we have

$$
\lim_{N^*\to\infty} \lim_{n\to\infty} \mathbb{E}(\boldsymbol{R}_{i,j}^2) = \frac{1}{n_1}\left(\sum_{k=1}^{\alpha} k^2 + \sum_{s=n_0+1+\alpha}^{n} s^2\right) = \frac{1}{n_1}\left(\tau(n) + \tau(\alpha) - \tau(\alpha+1+n_0)\right),
$$

where $\tau(\cdot)$ is the function of 'sum of squares', i.e., $\tau(n) = \sum_{k=1}^{n} k^2$.

For a malicious node, we have

$$
\lim_{N^*\to\infty} \lim_{n\to\infty} \mathbb{E}(\boldsymbol{R}_{i,j}^2) = \left(\alpha + \frac{1+n_0}{2}\right)^2,
$$

Therefore, according to Theorem 1,

$$
\begin{aligned}
\bar{s}_b^2 &= \lim_{N^*\to\infty} \lim_{n\to\infty} \lim_{p\to\infty} \frac{1}{p} \sum_{j=1}^{p} V_{bj} = \frac{1}{n_1}\left(\tau(n) + \tau(\alpha) - \tau(\alpha+1+n_0)\right) - \bar{\mu}_b^2, \\
\bar{s}_m^2 &= \lim_{N^*\to\infty} \lim_{n\to\infty} \lim_{p\to\infty} \frac{1}{p} \sum_{j=1}^{p} V_{mj} = 0.
\end{aligned}
$$

It completes the proof. $\qquad\square$

## F  NEURAL NETWORK CONFIGURATIONS

We train these models with a batch size of 10, an SGD optimizer operates with a learning rate of 0.01, and 0.5 momentum for 25 epochs. The accuracy of the model is evaluated on a holdout set of 1000 samples.

### F.1  FASHION-MNIST, MNIST AND QMNIST

- Layer 1: $1 * 16 * 5$, 2D Convolution, Batch Normalization, ReLU Activation, Max pooling.
- Layer 2: $16 * 32 * 5$, 2D Convolution, Batch Normalization, ReLU Activation, Max pooling.
- Output: 10 Classes, Linear.

### F.2  CIFAR-10

- Layer 1: $1 * 32 * 3$, 2D Convolution, Batch Normalization, ReLU Activation, Max pooling.
- Layer 2: $32 * 32 * 3$, 2D Convolution, Batch Normalization, ReLU Activation, Max pooling.
- Output: 10 Classes, Linear.

## G  METRICS

The metrics observed in Section 4 to evaluate the performance of the defense mechanisms are defined as follows:

$$\text{Precision} = \frac{\text{TP}}{\text{TP+FP}},$$
$$\text{Accuracy} = \frac{\text{TP+TN}}{\text{TP+FP+FN+TN}},$$
$$\text{Recall} = \frac{\text{TP}}{\text{TP+FN}},$$
$$\text{F1} = 2 \times \frac{\text{Precision} \times \text{Recall}}{\text{Precision+Recall}}.$$

## H MANDERA PERFORMANCE WITH DIFFERENT CLUSTERING ALGORITHMS

In this section, Figure 8 demonstrate that the discriminating performance of MANDERA when hierarchical clustering and Gaussian mixture models are used in-place of K-means for FASHION-MNIST data set remain robust.

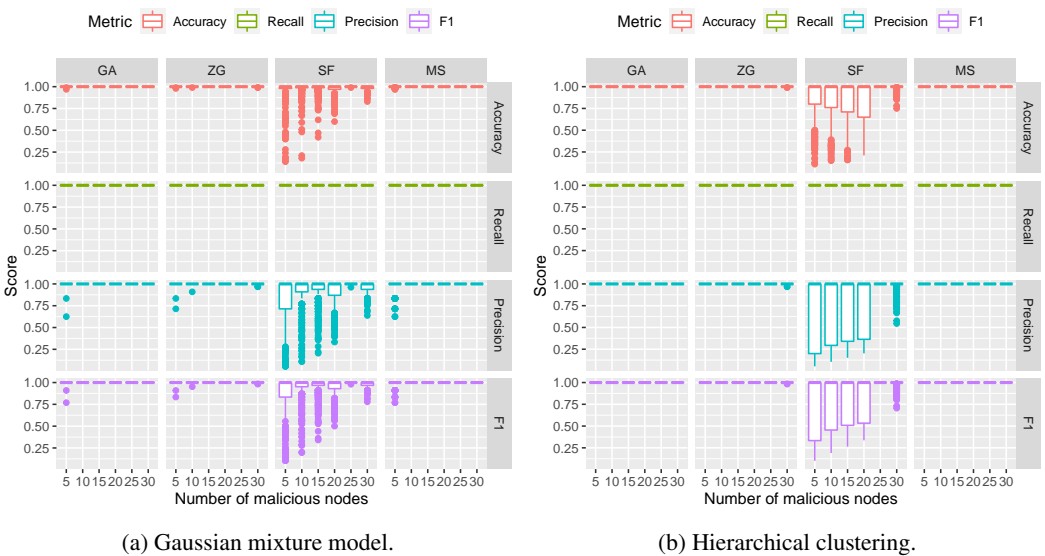

(a) Gaussian mixture model.          (b) Hierarchical clustering.

Figure 8: Classification performance of our proposed approach MANDERA (Algorithm 1) with other clustering algorithms under four types of attack for FASHION-MNIST data. GA: Gaussian attack; ZG: Zero-gradient attack; SF: Sign-flipping; and MS: mean shift attack. The boxplot bounds the 25th (Q1) and 75th (Q3) percentile, with the central line representing the 50th quantile (median). The end points of the whisker represent the Q1-1.5(Q3-Q1) and Q3+1.5(Q3-Q1) respectively.

## I MODEL LOSSES ON CIFAR-10, FASHION-MNIST AND MNIST DATA

Figure 9 presents the model loss to accompany the model prediction performance of Figure 6 previously seen in Section 4.

## J MODEL LOSSES ON QMNIST DATA

Figure 10 presents the model loss to accompany the model prediction performance of Figure 7b previously seen in Section 4.

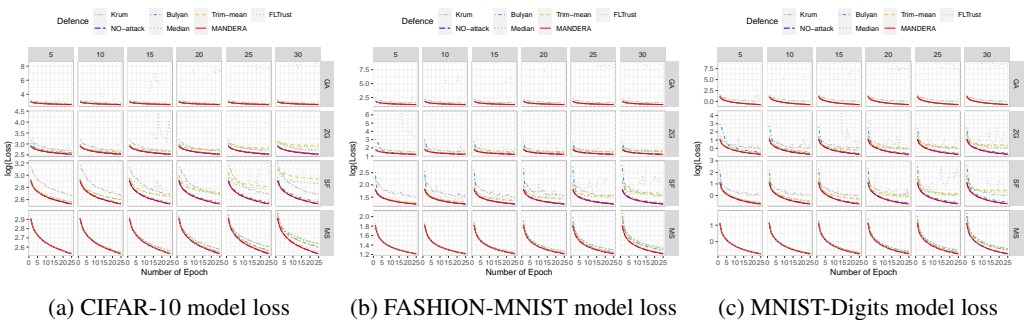

(a) CIFAR-10 model loss     (b) FASHION-MNIST model loss     (c) MNIST-Digits model loss

Figure 9: Model Loss at each epoch of training, each line of the curve represents a different defense against the Byzantine attacks.

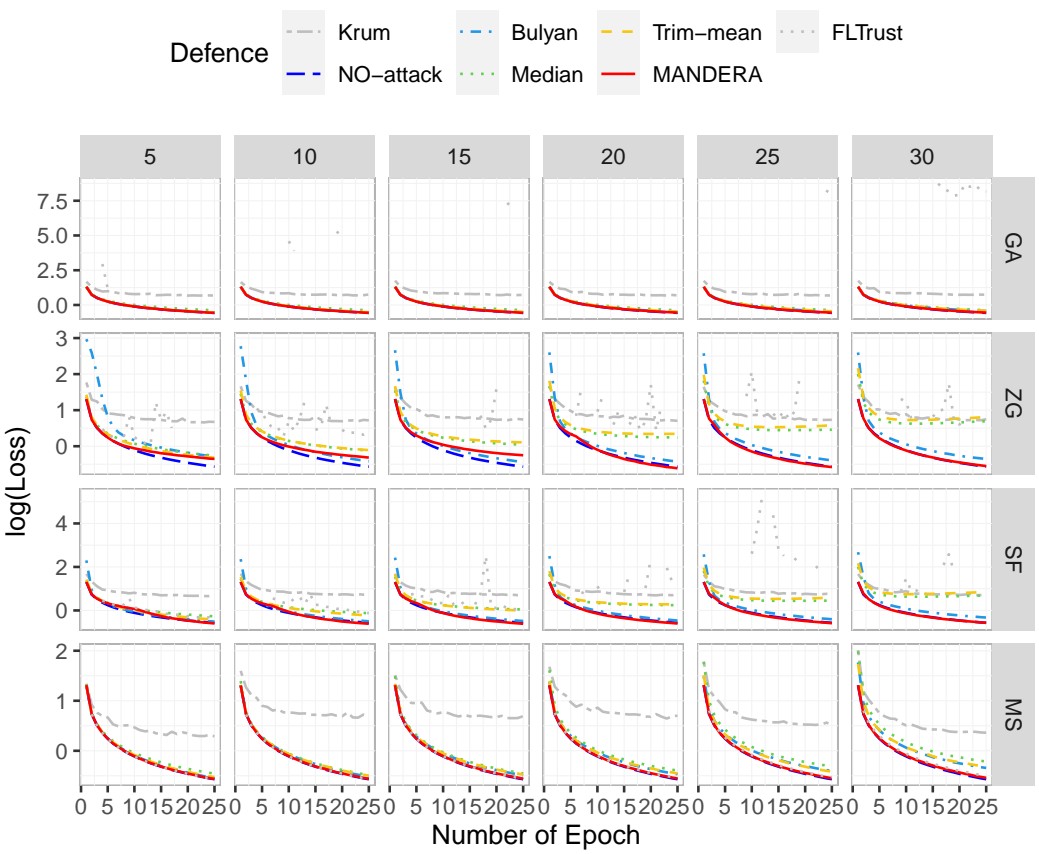

Figure 10: QMNIST model loss.

Figure 11: Model Loss at each epoch of training, each line of the curve represents a different defense against the Byzantine attacks.

## K    COMPUTATIONAL EFFICIENCY

We have previously been able to observe that MANDERA can perform at par with the current highest performing poisoning attack defenses. Another benefit arises with the simplification of the mitigation strategy with the introduction of ranking at the core of the algorithm. Sorting and Ranking algorithms are fast. Additionally, we only apply clustering on the two dimensions (mean and standard deviation of the ranking), in contrast to other works that seek to cluster on the entire node update Chen et al.

(2021). The times in Table 1 for MANDERA, Krum and Bulyan do not include the parameter/gradient aggregation step. These times were computed on 1 core of a Dual Xeon 14-core E5-2690, with 8 Gb of system RAM and a single Nvidia Tesla P100. Table 1 demonstrates that MANDERA is able to achieve a faster speed than that of single Krum [2] (by more than half) and Bulyan (by an order of magnitude).

Table 1: Mean and standard deviation of computational times for defense function given the same set of gradients from 100 nodes, of which 30 were malicious. Each function was repeated 100 times.

| Defense (Detection) | Mean ± SD (ms) | Defense (Aggregation) | Mean ± SD (ms) |
|---|---|---|---|
| *MANDERA* | 643 ± 8.646 | Trimmed Mean | 3.96 ± 0.41 |
| Krum (Single) | 1352 ± 10.09 | Median | 9.81 ± 3.88 |
| Bulyan | 27209 ± 233.4 | FLTrust | 361 ± 4.07 |

---

[2]The use of multi-krum would have yielded better protection (c.f. Section 4) at the behest of speed.

