# OpenReview forum: "MANDERA: Malicious Node Detection in Federated Learning via Ranking"
_ICLR.cc/2023/Conference — Submitted to ICLR 2023_

### Official Review · Reviewer_L24P · 2022-10-24

**Confidence:** 3
**Correctness:** 3
**Technical Novelty And Significance:** 2
**Empirical Novelty And Significance:** 2
**Recommendation:** 3

**Clarity, Quality, Novelty And Reproducibility:**

Clarity: The method is clear.
Quality: The quality is okay.
Novelty: The novelty is marginal.
Reproducibility: The authors provide the codebase for reproducing. I do not have the chance to run the provided code.


------------------------------------

Thanks for the rebuttal. I decide to keep the original score after reading it.

**Strength And Weaknesses:**

Strength:
1. The idea of converting numerical features to rank features is inspiring. Will combine them both further improve the detection accuracy?

Weakness:
1. The claim is too strong. "To address the above, for the first time, we propose MANDERA which is theoretically guaranteed to efficiently detect all malicious gradients under Byzantine attacks with no prior knowledge or history about the number of attacked nodes." I seriously doubt if it is theoretically possible to detect all the malicious nodes.
2. The main theorems are all asymptotic while in malicious nodes detection we care about non-asymptotic guarantee.
3. The IID assumption is a too strong for federated learning. It is well-known that data/gradients in federated systems are heterogeneous [1] and can be heavy-tailed with no-bounded second moments [2].

[1] FEDERATED OPTIMIZATION IN HETEROGENEOUS NETWORKS
[2] On the Heavy-Tailed Theory of Stochastic Gradient Descent for Deep Neural Networks


**Summary Of The Paper:**

The paper proposed a method to detect malicious nodes by converting the numerical features to rank features and then apply malicious nodes detection algorithm.

**Summary Of The Review:**

The overall quality of the paper is okay but the limitations I mentioned in the weakness section seem to be hard to solve. As a result, I tend to reject. I will re-score if the authors can fix any of the issues.

---

> ### Author Response · Authors · 2022-11-11
> **Response to Reviewer L24P**
>
> Thank you for your review and here is a point-to-point response to each of your concerns. We are happy to discuss any further concerns during the discussion period.
>
> * **Concern:** The claim is too strong. "To address the above, for the first time, we propose MANDERA which is theoretically guaranteed to efficiently detect all malicious gradients under Byzantine attacks with no prior knowledge or history about the number of attacked nodes." I seriously doubt if it is theoretically possible to detect all the malicious nodes.
>     + **Response:** A theoretic guarantee to detect malicious gradients is shown in Theorem 1. Theorem 1 shows that our algorithm can detect all the malicious nodes asymptotically, as long as the attacker has limited information on the benign distribution $F$. The full proof of this derivation can be found in Appendix A. We would like to modify our claim, if evidence showing that our claim is too strong can be provided.
> * **Concern:** The main theorems are all asymptotic while in malicious nodes detection we care about non-asymptotic guarantee.
>     + **Response:** We totally agree with you that the asymptotic results in Theorem 1 may have limitations. But, we also believe that these results indeed give us important insights to understand the problem and provide us useful tools for practical applications.
>     Although our results are derived when $N$, the sample size on each local node, and $p$, the number of parameters in the deep models, are large, we found that in most practical problems both $N$ and $p$ are often reasonably large to support the theoretical results to play a meaningful role. Our experiments confirmed this.
>
>         In addition, as far as we know, we are the first to provide asymptotic behaviours of benign and malicious nodes. This should be considered as a significant contribution to the field.
> * **Concern:** The IID assumption is a too strong for federated learning. It is well-known that data/gradients in federated systems are heterogeneous [1] and can be heavy-tailed with no-bounded second moments [2].
>     + **Response:**  Yes, we agree with you that the IID assumption is a strong assumption. In the current manuscript, we have shown by the experiments in section 4.2 that the proposed method is robust to violations of the IID assumption.

---

### Official Review · Reviewer_9NF6 · 2022-10-24

**Confidence:** 5
**Correctness:** 1
**Technical Novelty And Significance:** 1
**Empirical Novelty And Significance:** 1
**Recommendation:** 1

**Clarity, Quality, Novelty And Reproducibility:**

The paper is well-organized and easy to follow. The proposed defense is not applicable since it collapses under a well-crafted attack. The authors do not provide code for reproducibility.

**Strength And Weaknesses:**

Strength:

- The discussed topic of Byzantine robustness in federated learning is important.
- The paper is well-organized and easy to follow.

Weaknesses:

- (FATAL) Collapse under a particular attack. Let $g$ be the benign gradient and e be all one vector. Craft the byzantine gradients to be $g+\alpha e$, where $\alpha$ is an arbitrarily large number. Since $g+\alpha e$ and $g$ are the same in the ranking space, this attack can easily compromise the proposed MANDERA.
- Limited theoretical analysis. Only show the robustness of proposed MANDERA under some specific Byzantine attacks.

**Summary Of The Paper:**

In this paper, the authors propose MANDERA that detects Byzantine gradients in the ranking space. The authors analyze the robustness of proposed MANDERA under some specific Byzantine attacks. Experimental results verify the efficacy of proposed MANDERA.

**Summary Of The Review:**

Although the idea of detecting Byzantines in the ranking space is interesting, the proposed MANDERA fails under a particular attack.

---

> ### Author Response · Authors · 2022-11-11
> **Response to 9NF6**
>
> Thank you for your review. Here is a point-to-point response to each of your concerns. If there are any further concerns, we will be happy to discuss them with you during the discussion period.
>
> * **Concern:** (FATAL) Collapse under a particular attack. Let $g$ be the benign gradient and $e$ be all one vector. Craft the byzantine gradients to be $g+\alpha e$, where $\alpha$ is an arbitrarily large number. Since $g$ and $g+\alpha e$ are the same in the ranking space, this attack can easily compromise the proposed MANDERA.
>     + **Response:** This statement is not correct. Figure 2 shows the workflow of our algorithm. The vector $g$ in your comment should be one of the rows in matrix ${\boldsymbol{M}}$. Note that the operation of rank is conducted column-wise, not row-wise. Therefore, when $g$ is changed to $g+\alpha e$, the whole ranking matrix will be changed. Therefore, such an attack will be detected by our algorithm.
>     We hope you could re-check our algorithm's workflow and re-evaluate our work accordingly.
>
> * **Concern:** Limited theoretical analysis. Only show the robustness of proposed MANDERA under some specific Byzantine attacks.
>     + **Response:** Theorem 1 shows the theoretical analysis for general Byzantine attacks, and it does not require any specific restrictions regarding the Byzantine attacks. Section 3 showed theoretical results for more specific attacks.
>     To the best of our knowledge, the results reported in this study are the best theoretical results for general Byzantine attacks in the literature so far.

---

### Official Review · Reviewer_HdfK · 2022-10-24

**Confidence:** 4
**Clarity, Quality, Novelty And Reproducibility:** 1. The authors may put the illustrati…
**Correctness:** 4
**Technical Novelty And Significance:** 2
**Empirical Novelty And Significance:** 3
**Recommendation:** 3

**Strength And Weaknesses:**

Strength:
1. The proposed method is very effective in experiments with performance matching or even outperforming some existing methods (Figure 7b).
2. The analyses of ranking based method is valuable that provide insights on why MANDERA works well in high-dimensional or large datasets regime.

Weakness:
1. The considered attack is essentially not Byzantine attack as claimed, the distinguishable distribution assumption make the attack model much more restrictive in that there are attacks [1] that have similar distributions with benign nodes but can prevent the optimization process from convergence to optimum.
2. Another problem is how to measure the amount of distinguishability is not discussed. In what cases the clustering algorithms can detect those malicious node is not discussed.
3. Per the previous two points, there are gaps towards to the Byzantine robustness guarantees. We know that MANDERA is robust to some restricted class of attacks, but not general Byzantine attack, some attacks that fall into this class are analyzed. We know that asymptotically there will be two clusters, but we don't know in what cases these two clusters will be correctly labelled. In conclusion, there are interesting blocks in the analysis part, but the presented arguments are not enough to obtain any Byzantine robustness guarantees.
4. In Figure 7b, why there are only two methods compared for GA and MS cases? The proposed method is good, but it seems that it does not outperform Bulyan in all cases.

[1] Baruch, G., Baruch, M., & Goldberg, Y. (2019). A little is enough: Circumventing defenses for distributed learning. Advances in Neural Information Processing Systems, 32.

**Summary Of The Paper:**

This paper proposes a new ranking based method to detect Byzantine agents that participate in federated learning. MANDERA is a novel method that is intuitive and easy to implement, and show good performance in experiments. The authors also provide theoretical analyses to support that MANDERA is guaranteed to detect all Byzantine agents under certain conditions, and analyze 4 specialized attack instances.

**Summary Of The Review:**

This paper propose a novel method with good analysis blocks and effective experiments results, but from my perspective, it lacks in the theoretical arguments to guarantee Byzantine robustness as I detailed in "weakness" section. I think this paper can be improved towards a very good paper if more blocks are added, but the current form is limitedly persuasive.

---

> ### Author Response · Authors · 2022-11-11
> **Response to Reviewer HdfK**
>
> Thank you for your review and below is a point-to-point response to each of your concerns. We realize that some key points of our study have been misunderstood. We would be happy to discuss any further concerns during the discussion period.
>
>  *  **Question:** The considered attack is essentially not Byzantine attack as claimed, the distinguishable distribution assumption make the attack model much more restrictive in that there are attacks [1] that have similar distributions with benign nodes but can prevent the optimization process from convergence to optimum.
>
>       + **Response:** We have to clarify that the condition $F \neq G$, which is referred to as the "distinguishable distribution assumption" in your comment, is NOT an assumption, but a fact that naturally holds for all Byzantine attacks. Because the attackers, cannot precisely know the benign distribution $G$, since only finite samples of $G$ can be observed. Therefore, condition $F \neq G$ naturally holds for all Byzantine attacks, and thus we didn't restrict the attacker's behaviors in any way by such a condition.
>
>         In fact, if the condition does not hold, i.e., $F=G$, the malicious nodes controlled by the attacker would behave in the same way the benign nodes, making the attacker trapped in the dilemma that all attempted attacks fail because they are too futile. With such an insight, it is straightforward to see that Theorem 1 gives a very general theoretical result under a mild condition.
>
>         Our experiments, which cover the Mean-Shift (MS) attack studied in [1] and a few other attacks, have shown that the proposed method works very well for a wide range of Byzantine attacks. These results validate our Theorem 1 from the practical point of view, giving us confidence in the excellence of the proposed method in the non-asymptotic scenarios and scenarios where the malicious distribution $F$ is close to the benign distribution $G$.
>
>  *  **Question:** Another problem is how to measure the amount of distinguishability is not discussed. In what cases the clustering algorithms can detect those malicious node is not discussed.
>
>       + **Response:** As we mentioned above, we don't need to worry about how distinguishable $F$ and $G$ is, because as long as $F \neq G$, Theorem 1 shows that the proposed algorithm works asymptotically when $N$ and $p$ are large.
>
>  *  **Question:** Per the previous two points, there are gaps towards to the Byzantine robustness guarantees. We know that MANDERA is robust to some restricted class of attacks, but not general Byzantine attack, some attacks that fall into this class are analyzed. We know that asymptotically there will be two clusters, but we don't know in what cases these two clusters will be correctly labelled. In conclusion, there are interesting blocks in the analysis part, but the presented arguments are not enough to obtain any Byzantine robustness guarantees.
>
>      + **Response:** Based on our previous clarification, Theorem 1 covers a very general, instead of restrictive, collection of Byzantine attacks. MANDERA can detect the two clusters in a general Byzantine setting. In terms of clustering, we need to assume that the number of malicious nodes is less than that of benign nodes. So we can distinguish which group is malicious.
>  *  **Question:**  In Figure 7b, why there are only two methods compared for GA and MS cases? The proposed method is good, but it seems that it does not outperform Bulyan in all cases.
>      + **Response:**  This statement is not correct. Figure 7b compared 6 defence strategies under 4 different attacks, which shows that the proposed method outperforms or is as good as Bulyan in all cases.
>     We hope you could recheck the figure with more specific attention on this point. We hope you could reconsider your evaluation of our work accordingly.
>
> [1] Baruch, G., Baruch, M., \& Goldberg, Y. (2019). A little is enough: Circumventing defenses for distributed learning. Advances in Neural Information Processing Systems, 32.

---

> > ### Comment · Reviewer_HdfK · 2022-11-28
> > **Response to the rebuttal**
> >
> > I have carefully checked the figure 7b and found there are some more than two methods being compared, and I suggest zooming into the part over 80 and making the figure larger so audience don't need to zoom in too much to check the results.
> >
> > I think your arguments assume that the Byzantine attacks follow some parameterized distributions, which is not necessary true. A Byzantine attack can be adaptive to the states of the current system (thus dependent on the data distributions). However, you are right, in this type Byzantine attack, your arguments make absolute sense.
> >
> > I have no other questions and comments.

---

> > > ### Author Response · Authors · 2022-11-28
> > > **Response to the reply of Reviewer HdfK**
> > >
> > > Thanks for your reply. We will take care of figure 7b and make sure it is clear enough for the audience.
> > >
> > > It is much appreciated, if you would like to update your score to reflect your updated understanding of our paper.

---

### Author Response · Authors · 2022-11-24
**Rolling discussion**

Dear reviewers,

We hope you are all great. Although we have received negative comments, we have tried our best to address all of them. Can you have a look and see if you have any remaining concerns. We appreciate it very much if you could acknowledge our rebuttal.

Best wishes,
Authors.

---

### Decision · Program_Chairs · 2023-01-20

**Decision:**

Reject

**Justification For Why Not Higher Score:**

The paper presents a set of defenses for a limited attack scenario, based on questionable technical assumptions.

**Justification For Why Not Lower Score:**

N/A

**Metareview: Summary, Strengths And Weaknesses:**

The authors persent MANDERA  anew a defense against Byzantine attacks in federated learning settings. The authors provide some theoretical guarantees that their metheod can detect malicious gradients without prior knowledge or history of the number of attacked nodes. Their scheme operates by transforming the updating gradient matrix into a ranking matrix, which seems to allow to separate the malicious and non malicious gradients. To demonstrate experimental effectiveness, the authors provide  results agains some attack implementations (Gaussian, Zero Gradient, Sign Flipping, Shifted Mean) on IID and Non-IID datasets

The reviewers, identified some strengths, and several weaknesses in this work, as follows.

Strengths:
- The method is intuitive and easy to implement
- The performance in the settigns presented is good
- Combining numerical and rank features may further improve detection accuracy

however significant weaknesses were identified.

Weaknesses:
- The Attack setting studied is not a true Byzantine attack (e.g., not fully worst case)
- The assumption of distinguishable distributions restricts and IDD of data restricts the attack/defense model
- There are some technical gaps in theoretical arguments to guarantee Byzantine robustness and the overall theoretical claims are too strong, since the "guaranteed" property applies to a limited setting.

Although some Reviewers did not engage after the rebuttal with the paper, my own reading of the rebuttal, I agree with the criticisms, especially with regards to the generality of the statement. The authors mention that "Because the attackers, cannot precisely know the benign distribution G
, since only finite samples of G can be observed." This is indeed a limited setting, and not a truly worst case one, where the adversaries are omniscient. Limiting the scope of the attack is OK, but the authors seem to mischaracterize the generality of their setting. In fact it is quite unclear why the set of byzantine, and "true" gradients cannot have the same distribution, approximately, while hiding the attack at the "tail" of a distribution, that may be statistically impossible to use it for differentiation.



**Summary Of Ac-Reviewer Meeting:**

N/A